# Downregulation of S1P Lyase Improves Barrier Function in Human Cerebral Microvascular Endothelial Cells Following an Inflammatory Challenge

**DOI:** 10.3390/ijms21041240

**Published:** 2020-02-13

**Authors:** Bisera Stepanovska, Antonia I. Lange, Stephanie Schwalm, Josef Pfeilschifter, Sina M. Coldewey, Andrea Huwiler

**Affiliations:** 1Institute of Pharmacology, University of Bern, Inselspital, INO-F, CH-3010 Bern, Switzerland; bisera.stepanovska@pki.unibe.ch; 2Department of Anesthesiology and Intensive Care Medicine, Jena University Hospital, D-07747 Jena, Germany; antonia.lange@med.uni-jena.de (A.I.L.); Sina.Coldewey@med.uni-jena.de (S.M.C.); 3Septomics Research Center, Jena University Hospital, D-07747 Jena, Germany; 4Pharmazentrum Frankfurt/ZAFES, University Hospital, Goethe University Frankfurt am Main, Theodor-Stern Kai 7, D-60590 Frankfurt am Main, Germany; s.schwalm@med.uni-frankfurt.de (S.S.); Pfeilschifter@em.uni-frankfurt.de (J.P.); 5Center for Sepsis Control and Care, Jena University Hospital, D-07747 Jena, Germany

**Keywords:** blood–brain barrier, endothelial integrity, inflammation, S1P lyase, junctional molecules, PKC

## Abstract

Sphingosine 1-phosphate (S1P) is a key bioactive lipid that regulates a myriad of physiological and pathophysiological processes, including endothelial barrier function, vascular tone, vascular inflammation, and angiogenesis. Various S1P receptor subtypes have been suggested to be involved in the regulation of these processes, whereas the contribution of intracellular S1P (iS1P) through intracellular targets is little explored. In this study, we used the human cerebral microvascular endothelial cell line HCMEC/D3 to stably downregulate the S1P lyase (SPL-kd) and evaluate the consequences on endothelial barrier function and on the molecular factors that regulate barrier tightness under normal and inflammatory conditions. The results show that in SPL-kd cells, transendothelial electrical resistance, as a measure of barrier integrity, was regulated in a dual manner. SPL-kd cells had a delayed barrier build up, a shorter interval of a stable barrier, and, thereafter, a continuous breakdown. Contrariwise, a protection was seen from the rapid proinflammatory cytokine-mediated barrier breakdown. On the molecular level, SPL-kd caused an increased basal protein expression of the adherens junction molecules PECAM-1, VE-cadherin, and β-catenin, increased activity of the signaling kinases protein kinase C, AMP-dependent kinase, and p38-MAPK, but reduced protein expression of the transcription factor c-Jun. However, the only factors that were significantly reduced in TNFα/SPL-kd compared to TNFα/control cells, which could explain the observed protection, were VCAM-1, IL-6, MCP-1, and c-Jun. Furthermore, lipid profiling revealed that dihydro-S1P and S1P were strongly enhanced in TNFα-treated SPL-kd cells. In summary, our data suggest that SPL inhibition is a valid approach to dampenan inflammatory response and augmente barrier integrity during an inflammatory challenge.

## 1. Introduction

The blood–brain barrier (BBB) is a unique barrier present in all mammals and is localized at the interface between the blood and the central nervous system (CNS). It is an evolutionary trait crucial for the maintenance of CNS homeostasis, by regulating the exchange of blood-borne molecules and cellular elements [1,2]. This highly specialized structural and biochemical barrier is created by endothelial cells that line the microvessels and differs significantly from non-CNS endothelial cells [3]. The BBB’s properties are primarily determined by junctional complexes between the endothelial cells, which restrict the paracellular diffusion and preserve cell-cell contact [1]. BBB breakdown plays an important role in the pathogenesis of many CNS diseases, ranging from trauma to neurodevelopmental and neurodegenerative diseases, where proinflammatory substances or specific disease-associated proteins often mediate endothelial dysfunction [4,5]. Accordingly, restoring the barrier integrity is considered as a promising therapeutic approach, and several molecular pathways that regulate the BBB function and integrity have been described in this context [6]. One of the pathways that has been identified as a key determinant of BBB permeability is the sphingosine 1-phosphate (S1P) signaling cascade [7].

S1P is the central lipid molecule in the sphingolipid catabolism, which is generated by phosphorylation of sphingosine by the action of sphingosine kinase (SK)-1 and -2 [8]. The physiological importance of S1P is substantial, since it is ubiquitously distributed [9] and regulates a myriad of cellular responses, including survival, proliferation, migration, and differentiation [10]. S1P primarily acts as an extracellular high-affinity ligand and an activator of the five G protein-coupled receptors named S1P_1–5_ [11]. Signaling downstream of the S1P receptors modulates lymphocyte trafficking, vascular tone, vascular barrier function, cardiac development, and many other critical processes [12]. Furthermore, S1P can act as an intracellular second messenger to antagonize apoptotic signals and regulate the Ca^2+^ homeostasis [13]. However, these intracellular functions are not fully understood, and direct targets are still elusive.

To preserve the intracellular sphingolipid balance, which is known as “sphingolipid rheostat”, the cells exploit a series of enzymes that reversibly backconvert S1P to sphingosine, including lipid phosphate phosphatases and S1P phosphatases, or that irreversibly degrade S1P to hexadecenal and ethanolamine-phosphate and thereby terminate signaling. The latter is catalyzed by the endoplasmic reticulum-resident enzyme S1P lyase (*Sgpl1*, SPL) [14,15].

Several S1P receptors have been suggested to be involved in the development and maintenance of endothelial barrier properties. S1P in the blood at physiological concentrations activates S1P_1_ on the endothelial cells and induces barrier enhancement [7,16] by the modulation of cytoskeletal forces [17] and the redistribution of tight junction proteins [18]. S1P_5_ expressed on brain endothelial cells also contributes to optimal barrier formation [19], while an opposite function was described for S1P_2_, which induces stress fiber formation and prevents the proper localization of cell–cell junctions, resulting in enhanced barrier permeability [20]. S1P_3_ seems to act in a similar fashion as S1P_2_, as S1P_3_ inhibition functionally tightens the blood–tumor barrier in brain metastases [21], whereas S1P_3_ ligation, following intratracheal delivery of S1P, causes pulmonary edema via endothelial/epithelial barrier disruption [22]. Obviously, an important component of the mechanism of the action of extracellular S1P (eS1P) is the intercellular junctional complexes, which include the adherens and tight junction molecules and also several adhesion proteins outside of the specialized junctional complexes.

In contrast to the vast knowledge about the S1P receptor-mediated functions of eS1P, much less is known about the effects of the intracellular S1P (iS1P) in endothelial cells and, in particular, the impact on barrier function. In the present study, we used an SPL knockdown (kd) approach in endothelial cells derived from the BBB, HCMEC/D3, to evaluate the implications of increased intracellular S1P levels on endothelial phenotype markers, responses to inflammatory stimuli, and the consequences on barrier tightness. As the reports on SPL involvement in endothelial function are sparse and mainly derived from endothelial cells from non-CNS vascular beds, this is the first study that investigates the vasculoprotective role of SPL in the brain endothelium under inflammatory conditions.

In this study, we demonstrate that, in S1P lyase (SPL-kd) cells, transendothelial electrical resistance, as a measure of barrier integrity, was regulated in a dual manner. Unstimulated SPL-kd cells had a more destabilized barrier, whereas in an inflammatory setting, SPL-kd mediated protection from proinflammatory cytokine-mediated barrier breakdown. We show that SPL-kd resulted in a substantial accumulation of intracellular S1P and dihydro-S1P and various other molecular factors that could contribute to the protective effect. These data suggest that SPL modulation is a valid approach to dampen an inflammatory response and enhance barrier integrity during an inflammatory challenge in brain endothelial cells. Finally, they provide a clue how to treat diseases characterized by inflammation-mediated BBB disruption.

## 2. Results

Many studies have shown that S1P helps to maintain vascular endothelial barrier integrity which mainly occurs through S1P receptor signaling. In order to evaluate the effect of intracellular S1P on BBB function, we stably downregulated SPL in the human cerebral microvascular endothelial cell line HCMEC/D3 by transduction of cells with a lentiviral construct containing SPL-specific shRNA or an empty vector as a control. Knockdown efficiency was 99% on the SPL protein level (Figure 1A) and 95% on the mRNA level (Figure 1B). Interestingly, SPL-kd cells changed their morphology from a typical cobblestone shape to a rather spindle-shape phenotype (Figure 1C). To further characterize the SPL-kd cells, cellular S1P and dihydro-S1P were quantified by mass spectrometry. The results show an approx. 6-fold increase of iS1P in the SPL-kd cells as compared to the control cells (Figure 1D) and a small, but significant, change in the levels of dihydro-S1P (Figure 1E). When exogenous sphingosine was added to ensure sufficient substrate for SKs, endogenous S1P accumulated manifold and further increased by SPL-kd (Figure 1D).

In order to detect the differences in the endothelial barrier integrity of HCMEC/D3 control and SPL-kd cells, an electric cell-substrate impedance sensing (ECIS^TM^) assay was used. First, an initial cell number titration was performed to determine the optimal cell number and time frame of the assay.

Control HCMEC/D3 and SPL-kd cells were seeded with a density between 20,000 and 50,000 cells/mL to determine a cell concentration resulting in a stable, long-term barrier function. All cell densities of tested control cells showed a long-term, stable barrier function after 96 h (Figure 2). In contrast, SPL-kd cells reached a short barrier plateau (12 h duration) with a subsequent barrier breakdown (Figure 2). As a stable barrier function developed at a cell density of 50,000 cells/mL in both cell types during an overlapping time interval, this density was chosen for further experiments.

Proinflammatory factors, including the bacterial product lipopolysaccharide (LPS) and cytokines such as tumor necrosis factor (TNF)-α, interleukin (IL)-1β, IL-6, and interferon (IFN)-γ, are known to affect the stability of endothelial barriers. To elucidate the impact of inflammatory stimuli on HCMEC/D3 control and SPL-kd cells, we performed a dose-response experiment (1:100, 1:400, 1:800 and 1:1000) using ECIS^TM^ (Figure 3). After 24 h, we observed a significant decrease of resistance only in control cells induced by a dilution of 1:100 of LPS plus cytokine mix (LPS + Cyt), indicating an initial barrier breakdown, whereas both cell types showed no decline of resistance at LPS + Cyt dilutions of 1:400, 1:800, and 1:1000. At 120 h after administration of the inflammatory stimulus, we observed an on-going barrier breakdown in HCMEC/D3 cells by LPS + Cyt 1:100. However, also, the higher LPS + Cyt dilutions caused significantly extenuated barrier stability in HCMEC/D3 cells. As seen before, SPL-kd without LPS + Cyt could not maintain barrier stability on a long-term basis; only when applying inflammatory stimuli, SPL-kd significantly strengthened the endothelial barrier (Figure 3).

We further investigated whether SPL-kd cells are differentially equipped with junctional molecules, which could explain the observed phenotype in barrier integrity. The expression of various adherens junction molecules under resting and inflammatory conditions was evaluated. As seen in Figure 4 and Appendix A, SPL-kd cells showed an increased basal protein expression level of the platelet and endothelial cell adhesion molecule (PECAM-1) compared to the control cells, which is reduced upon stimulation with the proinflammatory cytokine TNFα but still remained at a much higher level in SPL-kd cells than in the control cells. Vascular endothelial cadherin (VE-cadherin) and β-catenin were also increased by SPL-kd but were not altered by TNFα stimulation. The levels of p120 catenin and the adaptor protein zonula occludens-1 (ZO-1) were not affected by SPL-kd or TNFα. The increased protein expression of PECAM-1 and VE-cadherin was in line with the increased mRNA expression by the SPL-kd (Appendix A), whereas β-catenin mRNA (Appendix A) was not significantly changed by SPL-kd or by TNFα. However, since one of the downstream targets of β-catenin, Axin2 [23], was upregulated by SPL-kd and reduced by TNFα on mRNA level (Appendix A), we conclude that the regulation of β-catenin by SPL-kd occurs on the post-translational level and has downstream consequences.

Furthermore, in endothelial cells, TNFα markedly induces the synthesis of intercellular adhesion molecule 1 (ICAM-1) and vascular cell adhesion molecule 1 (VCAM-1), which are both known to facilitate leukocyte transmigration [24]. Here, TNFα stimulation resulted in a strong induction of ICAM-1 (Figure 5A, Appendix A) and VCAM-1 protein (Figure 5A, Appendix A) as well as of their mRNA expressions (Figure 5B,C). We found that in SPL-kd cells, only VCAM-1 expression, but not ICAM-1, was significantly reduced by SPL-kd, suggesting a potential attenuation of leukocyte recruitment in SPL-kd. Moreover, gene expression of inflammatory products including monocyte chemoattractant protein 1 (MCP-1/CCL2), IL-6 and IL-8, were all induced by TNFα treatment, but only MCP-1 and IL-6 were significantly downregulated in SPL-kd cells (Figure 6A–C), whereas IL-8 expression was rather enhanced by SPL-kd (Figure 6D).

To further delineate the mechanism of the SPL-kd-mediated protection of barrier integrity under inflammatory conditions, we investigated various key signaling cascades and transcriptions factors. VCAM-1, MCP-1, IL-6, and IL-8 are all known to be critically regulated by the transcription factor nuclear factor κB (NFκB). In view of a previous report showing that intracellular S1P can directly bind to the TNF receptor-associated factor 2 (TRAF2) and trigger NFκB activation [25], we here determined whether NFκB was affected by SPL-kd. Phosphorylation of the p65 subunit of NFκB occurs upon TNFα stimulation and precedes the nuclear translocation and transcriptional regulation of target genes [26]. In SPL-kd cells, we did not observe any difference in TNFα-stimulated p65-NFκB phosphorylation compared to the control cells (Appendix A), thus excluding a link between iS1P and NFκB activation in this HCMEC system.

Two protein kinases that showed enhanced phosphorylation and activity by SPL-kd were p38-MAPK and the AMP-activated protein kinase (AMPK) (Figure 7A). Additionally, the total protein expression level of the transcription factor c-Jun, as well as its phosphorylation at Ser^63^ were reduced in SPL-kd, whereas JunB was not altered (Figure 7A). No significant changes in phospho-p42/p44-MAPK, phospho-Akt, and phospho-SAPK/JNK were observed by SPL-kd (Appendix A). Other transcription factors, including KLF2 and KLF4, which are considered as “molecular switches” regulating important aspects of vascular function and leukocyte adhesion [27,28], were found to be enhanced on the mRNA level but unchanged on the protein level (Appendix A). Interestingly, the protein kinase C (PKC) family was also found to be activated by SPL-kd. Since PKC consists of at least 11 isoenzymes, and so far single isoenzyme’s activities cannot be measured in cellular systems, we analyzed the total cellular PKC activity by determining the phosphorylation pattern of PKC substrates using a phospho-specific PKC substrate and a phospho-MARCKS antibody (Figure 7B). Notably, two bands of approx. 30 kDa and 40 kDa were clearly upregulated by SPL-kd and a similar pattern was observed for the major PKC substrate, MARCKS. These bands were also upregulated by short-term phorbol ester (TPA) stimulation.

A previous study reported that iS1P can directly inhibit histone deacetylases (HDACs) [29] and that, by such epigenetic interference, iS1P may regulate the expression of various genes. We therefore tested here whether SPL-kd affects overall cellular HDAC activity by using an in situ fluorometric assay and by detecting histone H3 lysine 3 acetylation, as the most reported residue connected to iS1P, by Western Blot analysis. However, in SPL-kd cells, no change in either overall cellular HDAC activity (Appendix A) or in acetylated histone H3 (Appendix A) as a target of HDAC1/2 occurred.

To further investigate whether certain sphingolipid species are differentially affected by TNFα treatment in SPL-kd cells, lipid extracts of cells were quantified by mass spectrometry. S1P levels, which were markedly enhanced in SPL-kd cells, were only slightly increased by TNFα treatment in control cells but showed a synergistic increase in TNFα plus SPL-kd (Figure 8A). Dihydro-S1P showed a similar regulation and was also synergistically increased by TNFα plus SPL-kd (Figure 8B). The precursor of these lipids, sphingosine, was also increased, whereas sphinganine and many ceramides and glycosylated ceramides were rather downregulated (Appendix A). This lipid pattern suggests a negative feedback in the de novo pathway with a potential impact on SK-1 expression, which is downregulated on both the protein and mRNA level (Appendix A), possibly as a mechanism to divert sphingosine from further S1P production.

Finally, to study the role of extracellular S1P on the barrier integrity of HCMEC/D3 control and SPL-kd cells, we performed a pre- and post-treatment with S1P (10 nM and 100 nM) using the ECIS^TM^ system to examine the ability of S1P to ameliorate barrier dysfunction induced by an inflammatory stimulus. Pre-treatment with S1P, given 4 h prior to LPS + Cyt stimulation, did not affect barrier stability in either cell types over the whole time period (Appendix A). Conversely, post-treatment with S1P, given 24 h after LPS + Cyt stimulation, significantly attenuated LPS + Cyt-induced barrier destabilization in control HCMEC/D3 cells, but this was only seen after 120 h (Appendix A). No effect of post-treatment with S1P was seen in SPL-kd cells (Appendix A) or with S1P alone in both cell lines (Appendix A).

## 3. Discussion

In the present study, we demonstrate that the genetic knockdown of SPL in the human cerebral microvascular endothelial cell line HCMEC/D3 alters barrier function, which occurs in a dual manner. Under inflammatory conditions, which normally trigger a continuous and long-lasting barrier breakdown (Figure 3), SPL-kd demonstrated consolidated barrier function. However, under unstimulated conditions of endothelial junctional assembly, SPL-kd cells had a delayed barrier build up, a shorter stable barrier plateau and thereafter a continuous breakdown (Figure 2). Obviously, SPL-kd, which primarily leads to enhanced iS1P levels, can have both barrier-protective and barrier-disruptive effects, depending on the milieu.

Increased endothelial permeability is considered a serious complication in many inflammatory diseases or as a factor that contributes to the pathology of neurodegenerative diseases, trauma, tumors, and ischemia. Such endothelial dysfunction may be induced by various factors, for instance, inflammatory cytokines, histamine, thrombin, LPS, vascular endothelial growth factor, and bradykinin [30,31,32]. Although these permeability-inducing factors bind to different receptors, their signaling converges mainly on the level of the adherens junction molecules VE-cadherin, β-catenin, and PECAM-1, which become disrupted due to downregulated expression or enforced internalization through phosphorylation, resulting in a diminished interaction with other junctional molecules and the actin cytoskeleton, reduced enrichment on cell-cell borders, and an overall decrease in the adhesion strength [30,31,33,34].

The protective effect of S1P on the endothelial barrier is well reported and has been confirmed in vitro in cell cultures of endothelial cells but also in vivo in the vasculature [35,36]. Upon S1P stimulation, there is a rapid strengthening of the barrier that involves various mechanisms, including Rac1 activation, phosphorylation of actin-binding proteins of the ezrin/radixin/moesin family, and the rearrangement of cytoskeletal proteins to the cell periphery [35,37]. Additionally, S1P can increase the expressions of VE-cadherin and PECAM-1 in human endothelial cells and decrease the phosphorylation-induced VE-cadherin/catenin complex destabilization [38,39,40]. These effects of eS1P are mediated by S1P receptors, of which most evidence is presented for S1P_1_ [38,41,42], but other receptor subtypes, such as S1P_3_ [43] and S1P_5_ [19], may also mediate protection. In opposition to this, a rather barrier disruptive function of S1P through S1P_2_ [20] and S1P_3_ [21,44] has also been described.

Our data now demonstrate that iS1P is also an important for the barrier function and affects the BBB, as seen in vitro in endothelial cells originating from BBB. SPL-kd, which results in increased iS1P (Figure 1D), has immediate consequences on the protein expression of various adherens junctions, signaling molecules, and transcription factors, which may promote barrier protection under inflammation. These include an increased expression of VE-cadherin, PECAM-1 and β-catenin (Figure 4), and an increased activation state of PKC, AMPK, and p38-MAPK but a reduced expression of the transcription factor c-Jun (Figure 7). Upon TNFα treatment, SPL-kd cells demonstrated a strong downregulation of the adhesion molecule VCAM-1 (Figure 5) and the cytokines IL-6 and MCP-1 (Figure 6), all of which could contribute to the protective effect under inflammatory conditions. In this view, it is known that VCAM-1 is upregulated by pro-inflammatory cytokines, and upon VCAM-1 cross-linking, which mimics clustering with integrins on leukocytes, signaling is induced that increases stress fiber formation and barrier opening [45,46]. IL-6 promotes a sustained loss of endothelial barrier function via JNK-mediated STAT3 phosphorylation and de novo synthesis [47], while MCP-1 (CCL2) acts in an autocrine manner on the endothelial CCR2 receptor to disrupt tight and adherens junctions [48,49]. Whether the increased IL-8 synthesis in SPL-kd cells has any clinical relevance is presently unclear, but remarkably, a previous study [50] also showed that a similar effect occurred in placentas of preeclampsia patients, which expressed less SPL and had higher IL-8 levels in the circulation [50].

Although these factors are classically regulated by NFκB, we could not see any change in NFκB activity in the SPL-kd (Appendix A), and therefore the mechanism of downregulation by SPL-kd remains unclear. Krüppel-like factors (KLF) are reported to inhibit the transcriptional activity of NF-κB by competing for its transcriptional coactivators and thereby down-regulating its expression of proinflammatory genes [51]. For instance, KLF2 is known as a key transcriptional regulator of endothelial proinflammatory activation, by the inhibition of the cytokine-mediated induction of VCAM-1 expression, without affecting the expression of ICAM-1 and, similarly, the depletion of KLF4 acted proinflammatory by the enhancement of TNFα-induced VCAM-1 expression [52,53]. Although we detected significantly higher mRNA levels of KLF2 and KLF4, their protein levels remained equal between control and SPL-kd cells (Appendix A). Therefore, the contribution of KLF2 and KLF4 for the observed anti-inflammatory response in the SPL-kd cells can be excluded. We also identified the transcription factor c-Jun to be significantly reduced in SPL-kd (Figure 7A). We therefore speculate that this reduction of c-Jun at least partially contributes to the observed barrier protection. This is in agreement with a previous study that specifically targeted c-Jun expression, which resulted in the suppression of vascular permeability and inflammation [54]. It is also likely that the diminished expression of c-Jun accounts for the lower expression of VCAM-1, as it was shown that the AP-1 complex mediates the effect of TNF-α in the regulation of VCAM-1 expression through the modulation of the NF-κB transactivation in endothelial cells [55]. In addition, p38-MAPK may independently or synergistically regulate the effects of TNFα, and it was demonstrated that the inhibition of this kinase suppressed the TNFα-induced expression of VCAM-1 without affecting ICAM-1 and that this stress kinase cascade is critical for the TNFα-induced expression of MCP-1 in endothelial cells [56,57]. This clearly opposes our findings, as we found a constitutive increase in p38-MAPK with a decreased basal production of MCP-1 and TNFα-induced VCAM-1, which corroborates the controversial role of the p38 pathway [58].

The barrier-protective effect of iS1P is in agreement with a previous study by Li et al. [59], who showed that mice transgenic for SK-1 in endothelial cells have a reduced vascular leakiness upon angiopoietin-1 treatment, and that this protective effect is independent of eS1P and S1P receptors. This was further confirmed by Zhao et al. [60] in another approach of targeting the degrading enzyme SPL. They showed that in human lung microvascular endothelial cells, or in heterozygous SPL (+/-) mice, the loss of SPL reduced LPS-triggered barrier breakdown and pulmonary permeability. The authors also suggested that the protection from lung endothelial barrier disruption is mediated by accumulated iS1P and requires coupling to S1P_1_ and subsequent Rac1 activation. Obviously, there is an inconsistency whether iS1P acts dependent on S1P receptors or not. Moreover, a study using intravital microscopy in a murine atherosclerosis model revealed that iS1P reduced leukocyte adhesion to capillary wall and decreased LPS-induced endothelial permeability [61], thus corroborating our findings.

In opposition to the restorative effect of SPL-kd in an inflammatory setting, our data show a barrier-disruptive effect of SPL-kd in the early phase of barrier build up. Biphasic barrier regulation has also been reported by other stimuli. In this regard, an androgen-like steroid, LPS, and cyclic nucleotides, demonstrated a dual manner of barrier regulation, which is concentration dependent and involves changes in the actin cytoskeleton, protein kinase A, and PI3K/Akt pathways among others [62,63,64,65]. Although SPL-kd cells possess enhanced expression of several factors important for barrier consolidation, these factors are not, per se, sufficient to support the barrier function in the initial phase of junctional assembly, and the mechanism of this is still unclear.

Various mechanisms can contribute to the basal assembly of endothelial junctions and among these are calcium and multiple protein kinases, including PKC, AMPK, protein kinase A, myosin light-chain kinase, non-receptor tyrosine kinases, and Rho-dependent kinase [66,67,68]. For instance, PKC and intracellular Ca^2+^ [69] were reported to promote junction assembly in MDCK cells, whereas already assembled junctions are destabilized by factors such as thrombin through PKC activation and increased intracellular Ca^2+^ [70]. Since SPL knockout in mouse embryonic fibroblasts (MEF) has been shown to enhance intracellular calcium levels [71,72], and since the inhibition of S1P synthesis by sphingosine kinase inhibition decreases Ca^2+^ mobilization and permeability in human umbilical vein endothelial cells [73], it is well conceivable that also in HCMECs, the downregulation of SPL causes changes in intracellular Ca^2+^ levels, which could regulate junctional assembly and disassembly. As conventional PKCs are usually activated by Ca^2+^, we studied the pattern of PKC substrates’ phosphorylation and MARCKS phosphorylation and could indeed see changes in PKC activity (Figure 7B). Notably, the phosphorylations of 30 kDa and 40 kDa substrates were increased in SPL-kd, suggesting that iS1P may either directly or indirectly activate PKC. This dual effect of PKC can be explained by the fact that PKC is not just one entity, but rather a family consisting of 11 different isoforms, which can exert diverse and even opposite functions. In this regard, PKC-α, PKC-β, -θ, and −ζ activities were attributed to increased endothelial permeability [74,75,76,77], whereas PKC-δ and PKC-ε can mediate barrier protection [78,79]. It is also worth noting that in the livers of SPL-deficient mice [80] and in MEFs isolated from SPL-deficient mice [81], diacylglycerols, which are well known direct PKC activators, were reported to be significantly increased, thus further supporting the conclusion that SPL-kd couples to enhanced PKC activity.

To examine the efficacy of S1P to ameliorate the barrier dysfunction induced by an inflammatory stimulus, we performed a pre- and post-treatment with S1P. Our data show that in control HCMEC/D3 cells, eS1P by itself, or eS1P added as a stimulus prior to an inflammatory mixture, could not strengthen the endothelial barrier. Only in an inflammatory setting, post-treatment with eS1P had a protective effect on the cytokine-induced barrier breakdown (Appendix A). Notably, this protective effect was seen at a late time point (120 h), but not at early time points (48 h). It may be speculated that this phenomenon is mediated by gene transcriptional mechanisms and de novo protein synthesis, which are triggered by eS1P and require a prolonged period until detection. The protective effect of post-treatment eS1P in the inflammatory setting was abolished in the SPL-kd cells (Appendix A), suggesting that either the expression of S1P receptors are altered by the knockdown and thus the responsiveness towards eS1P or that the barrier improving effect by SPL-kd is already maximal and cannot be further enhanced by additional eS1P. In this context, it is of interest that the barrier improving effect of eS1P cannot be seen in all types of endothelial cells. In a comparative study between dermal microvascular endothelial cells (HMEC-1) and glomerular endothelial cells (GENC), it was shown that eS1P could only improve the barrier function in HMEC-1, but not in GENC [82]. The protective effect was dependent on the ability of eS1P to stimulate AMPK, which was seen in HMEC-1 but not in GENC.

To see whether the protective effect of SPL-kd under inflammatory conditions correlates with a change in certain sphingolipid subspecies, a more extended analysis of sphingolipids was performed by LC-MS/MS (Figure 8 and Appendix A). Previous studies demonstrated that not only S1P, but also other sphingolipid molecules, such as sphingomyelin [83], ceramides, and very long-chain ceramides [84,85], are involved in endothelial barrier regulation. In agreement with this, other sphingolipid species may mediate the response towards inflammatory stimuli, as we have seen from the LC-MS/MS data that S1P and dihydro-S1P, which are both modestly increased by TNFα in control cells, synergistically increase by TNFα in SPL-kd (Figure 8). Ceramides of various chain lengths were decreased in HCMEC/D3 SPL-kd (Appendix A), which is in accordance to a previous study in SPL-deficient Hela cells [81]. According to Linder et al. [84], ceramides have the potential to alter endothelial cell permeability. Nevertheless, S1P and dihydro-S1P remain the only measured sphingolipids showing a unique pattern of increase in SPL-kd cells under inflammatory conditions. Since several studies have indicated a positive correlation between reduced dihydro-S1P and reduced endothelial barrier function [86], further investigation is required to elucidate to what extent dihydro-S1P is involved in the biphasic resistance response of SPL-kd.

Although most of the BBB characteristics are determined by endothelial cells, they coexist with pericytes and astrocytes and the surrounding extracellular matrix which maintain their characteristics [87]. Since our data were generated in immortalized brain endothelial cells, this bears the risk of undermining the barrier attributes that emanate from the continuous crosstalk of endothelial cells with mural and glial cells. Thus, it would be important to validate our findings in a co-culture system and expand our analysis on the tight junctions, which are crucial for BBB integrity.

Altogether, our data suggest that SPL inhibition is a valid approach to dampen an inflammatory response and increase barrier integrity during an inflammatory challenge in brain endothelium and it provides a clue how to treat diseases characterized by inflammation-mediated BBB disruption.

## 4. Materials and Methods

### 4.1. Chemicals

All chemicals, primer sequences, and commercial and in-house produced antibodies, are indicated in the Appendix A.

### 4.2. Cell Culturing and Stable SPL Knockdown Generation

Immortalized human cerebral microvascular endothelial cell line (HCMEC/D3) was purchased from CELLutions Biosystems Inc., Ontario, Canada (Catalogue Number: CLU512) and was maintained according to manufacturer’s instructions, with the variation of using DMEM as basal medium supplemented with 5% FBS from PAN-Biotech, 1.4 μM hydrocortisone, 5 μg/mL ascorbic acid, 5 mL CD lipid concentrate, 10 mM HEPES, and 1 ng/mL bFGF. Cells were passed 2–3 times a week, using Trypsin-EDTA 0.25% for dissociation, and were used within passages 27–40. Flasks were pre-coated with autoclaved 1% gelatin in phosphate-buffered saline (PBS) and incubated for 1 h in incubator to allow polymerization.

The stable knockdown of SPL in HCMECs was achieved by transduction with lentiviral short hairpin RNA (shRNA) construct from Sigma MISSION^®^ following the respective protocols. Two different constructs were tested in parallel (TRCN0000286832 and TRCN0000078314), and the clone that yielded strongest gene silencing (TRCN0000286832) was used further. Cells were further named SPL-kd. Virus control cells were generated with TRC2 pLKO.5-puro empty vector control plasmid DNA from the same company. For the selection of resistant colonies, 1.5 μg/mL of puromycin was added to the medium in accordance with the results from previous titration. Knockdown efficiency was confirmed by quantitative PCR, Western Blot analysis, and by an LC-MS/MS to determine intracellular S1P accumulation.

All cells were cultured in the environment of 37 °C in an atmosphere enriched with 5% CO_2_. Prior to stimulation, they were rendered serum-free for 4 h or 24 h with a medium consisting of DMEM, 0.1 mg/mL BSA, and 10 mM HEPES (later on referred as DMEM serum-free), unless otherwise stated.

### 4.3. Western Blot Analysis

Stimulated cells were washed with ice-cold PBS and were subsequently scraped with lysis buffer (50 mM Tris-HCl pH 7.4, 150 mM NaCl, 10% glycerol, 1% Triton X100, 2 mM EDTA pH 8.4, 2 mM EGTA pH 8.0, 40 mM β-glycerol phosphate, 50 mM NaF, 10 mM sodium pyrophosphate, 2 mM DTT, 200 μM Na_3_VO_4_, 400 μL reconstituted cOmplete™ protease inhibitor cocktail, and 10 μM PMSF). Cells were homogenized by sonication (5 s at 30 microns peak to peak amplitude; *n*. Zivy & Co Ltd., Oberwil, Switzerland), lysates were centrifuged for 10 min at 13,000 rpm, and the supernatant was taken for protein determination according to Bradford. The samples were dissolved in Laemmli buffer separated by SDS-PAGE followed by protein transfer to nitrocellulose membrane by wet blotting using a buffer containing 25 mM Tris, 190 mM glycine and 20% (v/v) methanol. Membranes were blocked with 3% (w/v) low-fat milk powder in PBS for 1 h and were then incubated with the respective antibodies and diluted in a buffer containing 50 mM Tris-HCl pH 7.4, 200 mM NaCl, 10% (v/v) horse serum, 3% (w/v) BSA fraction V, and 0.1% (*v*/*v*) Tween20.

### 4.4. RNA Extraction and Quantitative PCR Analysis

Stimulated cells were washed with ice-cold PBS and homogenized in RNA-Solv^®^ reagent. Total RNA extraction was performed according to the instructions of the manufacturer. The yield and purity of the isolates were assessed with a NanoDrop^®^ ND-1000 spectrophotometer (Witec AG, Littau, Switzerland), and first strand cDNA was synthesized using 1 μg total RNA as template. SYBR^®^ Green-based quantitative PCR was performed in a BioRad CFX Connect™ Optics Module thermal cycler (Bio-Rad Laboratories Inc., Hercules, CA, USA). The Bio-Rad CFX Manager software was used to monitor the melting curve, and to obtain the quantification data. The relative mRNA expression of the gene of interest was calculated with the ∆∆*C*t method normalized to 18S RNA as a housekeeping gene.

### 4.5. MCP-1 ELISA

A human MCP-1 PicoKine™ ELISA kit (Boster Biological Technology, Pleasanton, CA, USA) was used to quantify MCP-1 in cell culture supernatants according to the manufacturer’s instructions. Confluent cells in 100 mm-diameter dishes were incubated for 4 h in DMEM serum-free prior to stimulation with 1 nM TNFα. Cell supernatants were collected, centrifuged for 5 min at 14,000× *g* and 4 °C, and the supernatant was used in a 1:100 dilution.

### 4.6. Quantification of Sphingolipids by LC-MS/MS

Cell monolayers in 60 mm-diameter dishes were trypsinized, pelleted, and resuspended in methanol containing C17-sphingolipids as internal standards and were subjected to lipid extraction and LC-MS/MS analysis, as previously described [88].

### 4.7. Barrier Integrity Measurements Using the ECIS^TM^ System and Inflammatory Stimulus Preparation

The endothelial barrier integrity of the confluent HCMEC/D3 control and SPL-kd monolayers was assessed using an electric cell-substrate impedance sensing (ECIS^TM^) Zθ system (Applied Biophysics Inc., Troy, NY, USA). ECIS is a real-time and impedance-based method to study the barrier dynamics of cells grown onto gold-filmed wells. Semi-confluent HCMEC/D3 control and SPL-kd cells (passage 44–46) were seeded on 96W10idf ECIS Cultureware^TM^ Arrays (Applied Biophysics Inc., Troy, NY, USA), which were pre-coated with human fibronectin (1 mg/mL) and murine collagen IV (1.25 mg/mL) in Hank’s balanced salt solution for 30–60 min. Partial media changes were performed every 24 h until a steady state of resistance (approx. 1100–1400 Ω) was reached, corresponding with the development of a confluent monolayer by the time stimulation experiments were commenced. ECIS experiments were continuously monitored for 5–6 days to capture both acute and longer-term changes in the endothelial resistance. The impedance of cell-covered electrodes was measured with the multiple frequencies over time (MFT) mode to record the impedance measurements over a broad spectrum of frequencies. The resistance represents the integrity of cell barriers and was monitored at 4000 Hz. Each measurement was performed in triplicates. The time was resampled to 600 s and the medium control was subtracted from the mean value of each measurement. In the barrier integrity experiments, an inflammatory stimulus was used. The stock solution of the inflammatory stimulus (LPS + Cyt) consisted of lipopolysaccharide (LPS, 10 µg/mL, *E. coli* O111:B4) and the pro-inflammatory cytokines TNF-α (5 µg/mL), IL-1β (1 µg/mL), IFN-γ (1 µg/mL), and IL-6 (1 µg/mL) in DMEM/1% (w/v) BSA. The cells were incubated with either 1:100, 1:400, 1:800, or 1:1000 LPS + Cyt dilutions. S1P was solubilized in methanol and applied to cells in a final concentration of 10 nM or 100 nM as pre- (4 h before LPS + Cyt) or post-treatment (24 h after LPS + Cyt).

### 4.8. Statistical Analysis

Statistical analysis was performed by one-way ANOVA or an unpaired *t*-test where applicable. For multiple comparisons, the level of significance was calculated with Bonferroni correction. GraphPad Prism Software, version 6, (San Diego, CA, USA) was used for statistical analysis and graph presentations. For ECIS™ resistance curves, mean values and standard deviations were plotted against time starting after LPS + Cyt or S1P stimulation. The statistics were verified as two-way ANOVA with α = 0.05 and Bonferroni’s multiple comparisons test. The data are depicted as means ± S.D. for *n* number of replicates and are representatives from 3 independent experiments.

## Figures and Tables

**Figure 1 ijms-21-01240-f001:**
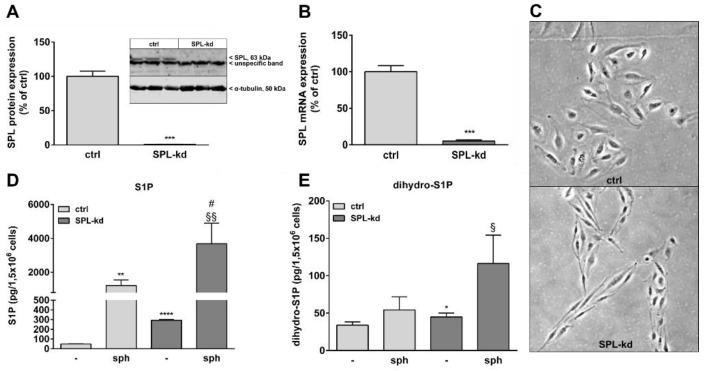
Characterization of a stable SPL knockdown in HCMEC/D3 cells. HCMEC/D3 cells, transduced with either an empty lentiviral vector (ctrl) or a lentiviral vector containing SPL shRNA (SPL-kd), were cultured until confluence and were then incubated for 4 h in serum-free DMEM. Proteins and RNA were extracted and taken for either a Western Blot analysis of the SPL protein (**A**), or a qPCR analysis of SPL mRNA (**B**). (**C**) Subconfluent cells grown on gelatin-coated dishes were photographed using a light microscope (Zeiss AxioObserver Z1, Feldbach) with a 200× total magnification and phase contrast setting. (**D**,**E**) Control (ctrl) and SPL-kd cells were rendered serum-free for 24 h and were treated for the last 10 min with either vehicle (-) or 1 μM of sphingosine (sph). The lipids were then extracted and processed for LC-MS/MS as described in the Methods section. The results in A and B are expressed as % of the control transduced cells and are depicted as means ± S.D. (*n* = 3 in A, *n* = 4 in B, *** *p* < 0.001). Results in D and E are expressed as pg/1,5 × 10^6^ cells and are means ± S.D. (*n* = 3; * *p* < 0.05, ** *p* < 0.01, **** *p* < 0.0001 considered statistically significant when compared to the vehicle-treated control; ^#^
*p* < 0.05 compared to the sphingosine-treated control; ^§^
*p* < 0.05, ^§§^
*p* < 0.01 compared to the vehicle-treated SPL-kd).

**Figure 2 ijms-21-01240-f002:**
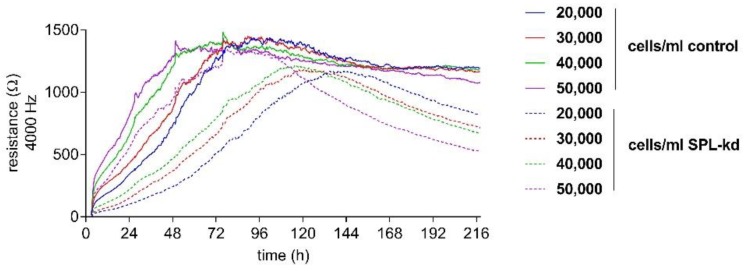
Cell number titration of HCMEC/D3 control and SPL-kd using ECIS^TM^. HCMEC/D3 control cells (continuous lines) and SPL-kd cells (dashed lines) were seeded at densities between 20,000 and 50,000 cells/mL. ECIS^TM^ measurements were monitored over an observation period of 216 h and were performed as described in detail in the Methods section. Partial medium changes were performed every 24 h until t = 72 h. The data are shown as mean curves of triplicate samples.

**Figure 3 ijms-21-01240-f003:**
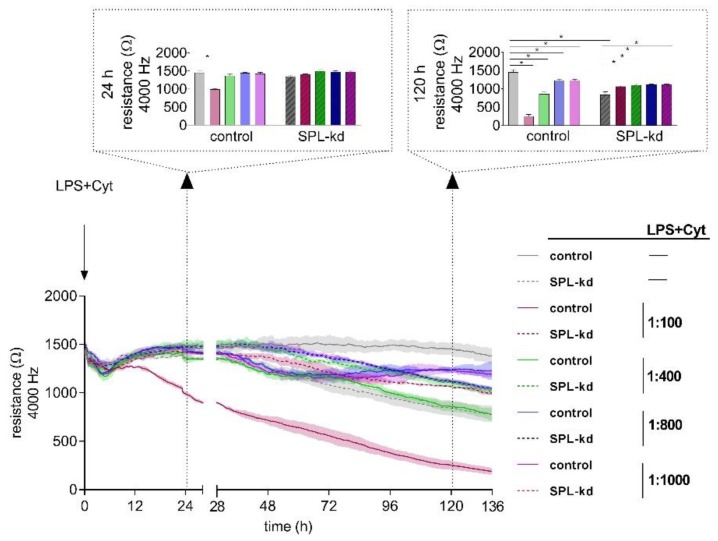
Impact of an inflammatory stimulus (lipopolysaccharide (LPS) + Cyt) on the barrier integrity of HCMEC/D3 control and SPL-kd cells. After the development of a stable barrier (t = 0 h), HCMEC/D3 control cells (continuous line) and SPL-kd cells (dashed line) were stimulated with different dilutions of an inflammatory stimulus (LPS + Cyt). Resistance values were analyzed at the two observation time points 24 h and 120 h (indicated by dotted arrows) after LPS + Cyt administration. The data are expressed as a mean ± S.D. (*n* = 3), * *p* < 0.05.

**Figure 4 ijms-21-01240-f004:**
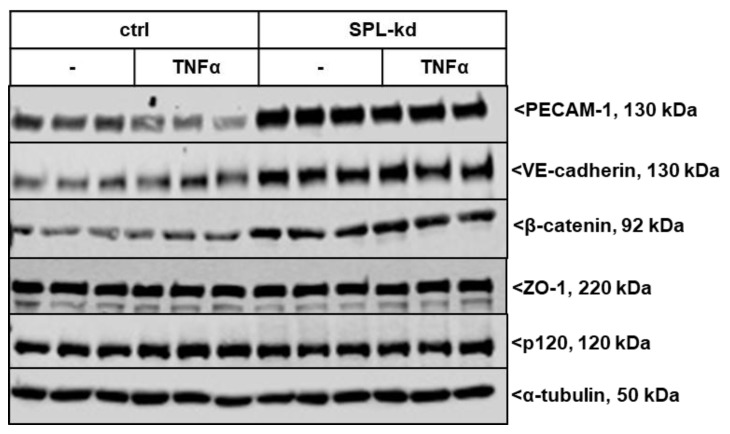
Effect of SPL knockdown on the expression of adherens junction molecules in TNFα-stimulated HCMECs. Confluent control (ctrl) and SPL-kd HCMEC/D3 cells were incubated for 4 h in serum-free DMEM before stimulation for 24 h with either vehicle (-) or 1 nM TNFα in DMEM/0.1% FBS. The proteins were extracted, separated by SDS-PAGE, transferred to nitrocellulose membrane and subjected to analysis using antibodies against PECAM-1, VE-cadherin, β-catenin, p120, ZO-1 and α-tubulin. Data show representative blots, out of 3 independent experiments, performed in triplicates. The evaluation of the respective bands is presented in Appendix A.

**Figure 5 ijms-21-01240-f005:**
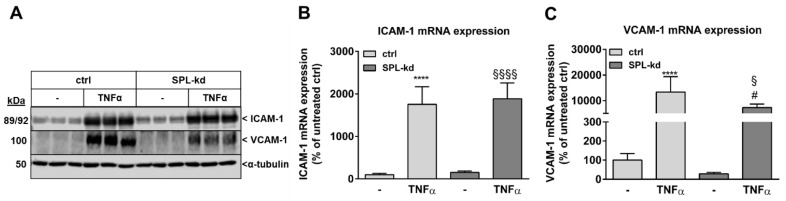
Effect of TNFα stimulation on the protein and mRNA expression of adhesion molecules in control and SPL-kd HCMEC/D3. Confluent control (ctrl) and SPL-kd HCMEC/D3 cells were rendered serum-free for 4 h prior to stimulation for 24 h with either vehicle (−) or 1 nM TNFα in DMEM/0.1% FBS. Thereafter, cells were taken for either protein extraction and Western Blot analysis of ICAM-1, VCAM-1, and α-tubulin (**A**), or RNA extraction and qPCR analysis of ICAM-1 and VCAM-1 (**B**,**C**). The results are expressed as % of control transduced cells and are means ± S.D. The evaluation of the respective bands (**A**) is presented in Appendix A. (*n* = 3 for A, *n* = 4–6 for B–C; **** *p* < 0.0001 compared to the vehicle-treated control; ^#^
*p* < 0.05 compared to the TNFα-treated control; ^§^
*p* < 0.05, ^§§§§^
*p* < 0.0001 compared to the vehicle-treated SPL-kd).

**Figure 6 ijms-21-01240-f006:**
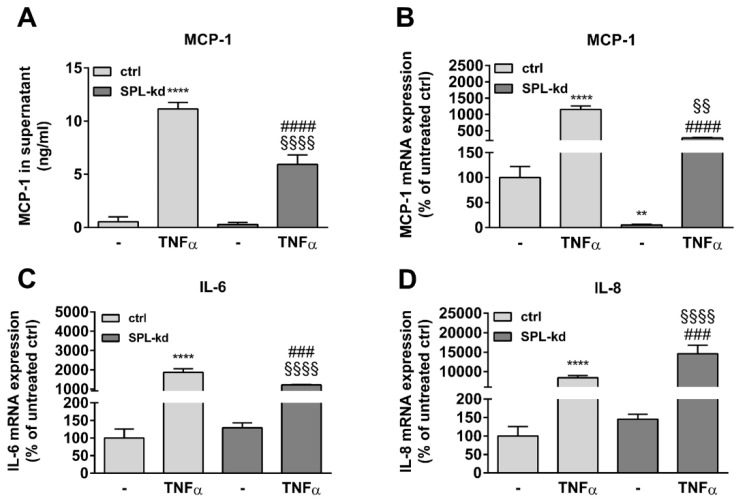
Effect of TNFα treatment on MCP-1, IL-6, and IL-8 expression in control and SPL-kd HCMEC/D3. A human MCP-1 ELISA kit was used to quantify secreted MCP-1 from vehicle and TNFα-stimulated HCMEC/D3 (**A**). (**B**–**D**): cells stimulated for 24 h with either vehicle (−) or 1 nM TNFα in DMEM/0.1% FBS were taken for RNA extraction and qPCR analysis using primers for MCP-1 (**B**), IL-6 (**C**), and IL-8 (**D**). Results in A are expressed as ng/mL MCP-1 in the supernatant and are means ± S.D (*n* = 3). The results in B–D are expressed as % of vehicle-treated control cells and are means ± S.D. (*n* = 3); ** *p* < 0.01, **** *p* < 0.0001 compared to the vehicle-treated control; ^###^
*p* < 0.001, ^####^
*p* < 0.0001 compared to the TNFα-treated control; ^§§^
*p* < 0.01, ^§§§§^
*p* < 0.0001 compared to the vehicle-treated SPL-kd.

**Figure 7 ijms-21-01240-f007:**
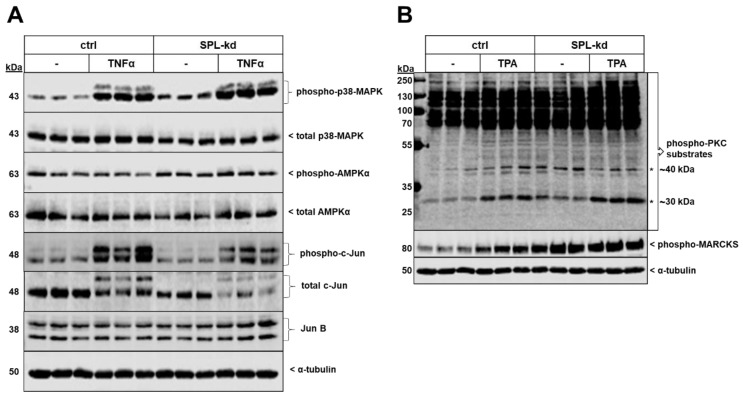
Effect of SPL-kd on various signaling molecules and transcription factors. Confluent control (ctrl) or SPL-kd HCMEC/D3 cells were rendered serum-free for 4 h prior to stimulation with either vehicle (−), 1 nM TNFα (**A**) or 50 nM TPA (**B**) in serum-free DMEM for 10 min. Protein extracts were separated by SDS-PAGE and subjected to Western blot analysis using antibodies against phospho-p38-MAPK, total p38-MAPK, phospho-AMPKα, total AMPKα, phospho-c-Jun, total c-Jun, JunB, phospho-protein kinase C (PKC) substrates, phospho-MARCKS, and α-tubulin. The data show representative blots, out of 3–4 independent experiments, performed in triplicates.

**Figure 8 ijms-21-01240-f008:**
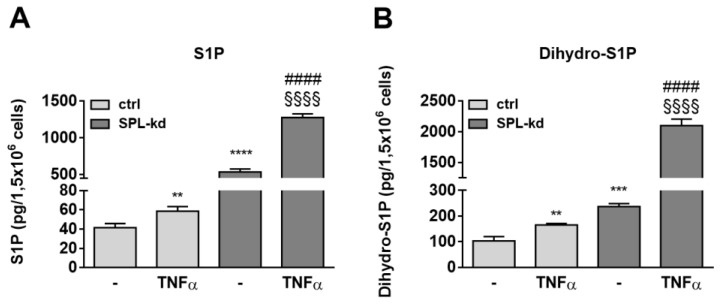
Effect of TNFα on S1P (**A**) and dihydro-S1P (**B**) content in HCMEC/D3 cells. Confluent control (ctrl) and SPL-kd cells were rendered serum-free for 24 h prior to stimulation for 24 h with either vehicle (−) or 1 nM TNFα. Lipids were extracted and quantified by LC-MS/MS. The results are depicted as picograms per 1.5 × 106 cells and are means ± S.D. (n = 3; * *p* < 0.01, ** *p* < 0.001, **** *p* < 0.0001 considered statistically significant when compared to the vehicle-treated control; #### *p* < 0.0001 compared to the TNFα -treated control; §§§§ *p* < 0.01 compared to the vehicle-treated SPL-kd).

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
