# Peer review of "Downregulation of S1P Lyase Improves Barrier Function in Human Cerebral Microvascular Endothelial Cells Following an Inflammatory Challenge"

_ijms, 2020, doi:10.3390/ijms21041240_

Round 1

Reviewer 1 Report

Dear authors

The manuscript entitled "Downregulation of S1P lyase mediates an improved barrier function in human cerebral microvascular endothelial cells following an inflammatory challenge" has presented interesting data about the intracellular S1P in endothelial cells, and in particular the impact on the barrier function. Truly, the effects of intracellular S1P in the endothelial cells were seldom discussed in published articles. Your experimental design is mainly in vitro, using S1P lyase knockdown approach in endothelial cells (Blood-Brain Barrier hCMEC/D3 cell line) to evaluate the consequences on endothelial barrier function and on molecular factors that regulate barrier tightness under normal and inflammatory conditions. The novelty of this manuscript is on the exploration between intracellular S1P, S1P lyase, and Blood-Brain Barrier. It's fascinating, but I have a few questions and recommendation on your current manuscript. My questions are the followings.   1. Your results and conclusion were basically based on in vitro experiments. If you could have animal models or data from human tissues, it will be much more fascinating. Would the authors possibly provide in vivo data?   2. The authors should make a graph summary to simply illustrate their hypothesis and possible mechanism based on the experimental results.   3. The authors used Electric Cell-substrate Impedance Sensing, known as ECIS, as a system for the measurement of transendothelial electrical resistance, as a measure of barrier integrity. It's a widely accepted electrical parameter to assess barrier integrity and suitability of in vitro cellular barriers for transport studies. However, it consists of only mono-layered endothelial cells. The blood-brain barrier consists of endothelial cells, astrocytes, and pericytes which make it a unique barrier. The authors should discuss their limitation, potential bias, and probably further experiments on the tight junctions between endothelial cells, astrocytes, and pericytes.   4. In Figure 5 and results, the authors firstly said that "TNF alpha stimulation resulted in a strong induction of ICAM-1 and VCAM-1 protein (Fig. 5A) as well as of their mRNA expressions (Fig. 5B and C)." However, I didn't be convinced by the dim expression of VCAM-1 protein in the western blotting (Fig. 5A). Especially, the VCAM-1 protein expression was much stronger in the control group with TNF-alpha pretreatment than in the SPL-kd group with TNF-alpha pretreatment. The data in Fig 5A was different from the dramatic differences of VCAM-1 mRNA between the two groups. Moreover, the authors even explained "We found that in SPL-kd cells, only VCAM-1 expression, but not ICAM-1, was significantly reduced by SPL-kd, suggesting a potential attenuation of leukocyte recruitment in SPL-kd." Although the authors had explained a few reasons in the discussion, I think they should explain more why VCAM-1 protein expression was much lower in SPL-kd cells compared to control, but they just further hypothesized a potential down regulation of leukocyte recruitment.   5. Upon TNF-alpha treatment, SPL-kd cells demonstrated a down regulation of the adhesion molecule VCAM-1, and the cytokines IL-6 and MCP-1, but the IL-8 expression was rather accelerated by SPL-kd cells. What may be the possible reasons toward the different expressions, especially they were known to be possibly associated? 

Author Response

1. Your results and conclusion were basically based on in vitro experiments. If you could have animal models or data from human tissues, it will be much more fascinating. Would the authors possibly provide in vivo data?

It would indeed be exciting to translate our findings in vivo and confirm that within the inflammatory milieu of multiple sclerosis or other neuroinflammatory processes, SPL inhibition strengthens the BBB and decreases the leukocyte transmigration potential. Notably, very few studies already point in this direction. Thus, inducible SPL knockout mice, or humanized mice with diminished SPL activity, show a protection in an EAE model and a cerebral malaria model (Billich et al., 2013; Finney et al., 2011), and the application of a SPL inhibitor also  reduced disease symptoms in the EAE model (Weiler et al., 2014) and in cerebral malaria (Finney et al., 2011). In these studies, alterations in BBB were not directly approached.

2. The authors should make a graph summary to simply illustrate their hypothesis and possible mechanism based on the experimental results.  

As suggested, we now included a summarizing graph as graphical abstract.

 3. The authors used Electric Cell-substrate Impedance Sensing, known as ECIS, as a system for the measurement of transendothelial electrical resistance, as a measure of barrier integrity. It's a widely accepted electrical parameter to assess barrier integrity and suitability of in vitro cellular barriers for transport studies. However, it consists of only mono-layered endothelial cells. The blood-brain barrier consists of endothelial cells, astrocytes, and pericytes which make it a unique barrier. The authors should discuss their limitation, potential bias, and probably further experiments on the tight junctions between endothelial cells, astrocytes, and pericytes.  

As suggested, a section about the limitations, potential bias, and further experiments was included in the manuscript (lines 393-398).

4. In Figure 5 and results, the authors firstly said that "TNF alpha stimulation resulted in a strong induction of ICAM-1 and VCAM-1 protein (Fig. 5A) as well as of their mRNA expressions (Fig. 5B and C)." However, I didn't be convinced by the dim expression of VCAM-1 protein in the western blotting (Fig. 5A). Especially, the VCAM-1 protein expression was much stronger in the control group with TNF-alpha pretreatment than in the SPL-kd group with TNF-alpha pretreatment. The data in Fig 5A was different from the dramatic differences of VCAM-1 mRNA between the two groups. Moreover, the authors even explained "We found that in SPL-kd cells, only VCAM-1 expression, but not ICAM-1, was significantly reduced by SPL-kd, suggesting a potential attenuation of leukocyte recruitment in SPL-kd." Although the authors had explained a few reasons in the discussion, I think they should explain more why VCAM-1 protein expression was much lower in SPL-kd cells compared to control, but they just further hypothesized a potential down regulation of leukocyte recruitment.  

A better quality blot of VCAM is now included in Fig. 5A, and we would like to stress that the densitometric evaluation of the VCAM band revealed a 15-fold induction in the control group as compared to 7-fold induction in the SPL-kd group upon TNFα stimulation. We have now also integrated a paragraph in the discussion where we disclose several findings from the literature that support the altered expression of VCAM-1 (lines 307-310 and 316-324).

5. Upon TNF-alpha treatment, SPL-kd cells demonstrated a down regulation of the adhesion molecule VCAM-1, and the cytokines IL-6 and MCP-1, but the IL-8 expression was rather accelerated by SPL-kd cells. What may be the possible reasons toward the different expressions, especially they were known to be possibly associated? 

Indeed, SPL-kd cells show a very peculiar pattern of inflammatory mediators’ production and while some are downregulated, others, like IL-8, are upregulated already on basal level. Although IL-6 and IL-8 are both considered pro-inflammatory, their regulation is not identical. Regarding IL-8, it was previously shown by Yang et al. (2016) that in placentas derived from preeclampsia patients, SPL was reduced and correlated with increased IL-8 in the circulation. This is now included in the discussion section (line 300-302).

Reviewer 2 Report

The manuscript is an interesting study addressing the role of S1P signaling in endothelial integrity in the CNS. It is a plus that the study explores in depth the signaling involved. However, the authors should use caution when extrapolating the conclusions beyond the in vitro system and the shRNA approach, especially because only one clone expressing the S1P lyase shRNA was used. Several issues need to address before the manuscript is accepted for publication in the IJMS.

Specifically:

In Fig.1E sphingosine was shown to increase the levels of dyhydroS1P. Since sphingosine cannot be converted to dihydrosphingosine, this result requires an explanation of the broader effect of S1P lyase down-regulation on sphingolipid metabolism. Are any of the SKs up-regulated in the SPL-kd cells?

Similarly, TNFalpha treatment of SPL-kd cells leads the up-regulation of sphinganine/dihydrosphingosine, the precursor of dihydroS1P (suppl. table). This could only be explained by an effect of TNFalpha on the de-novo sphingolipid metabolism independent of S1P lyase downregulation.

There is no evidence provided that S1P and dihydroS1P were not secreted when overproduced in the SPL-kd cells or as a result of sphingosine or TNFalpha treatments, therefore the conclusion that it is only intracellular effects of S1P or possibly dihydroS1P is incorrect. The S1P has been suggested to have an autocrine effects, which was not discussed as a possibility in this manuscript. Labels on the blots of figs. 4, 5, and 7 need to be improved; It is not clear what the individual bands represent. Are those technical repeats and if yes of what? It is also not clear which bands are control bands and which are TNFalpha or TPA treated. This makes it difficult to interpret the results.

Author Response

In Fig.1E sphingosine was shown to increase the levels of dyhydroS1P. Since sphingosine cannot be converted to dihydrosphingosine, this result requires an explanation of the broader effect of S1P lyase down-regulation on sphingolipid metabolism. Are any of the SKs up-regulated in the SPL-kd cells?

One possible explanation for the increased dihydro-S1P by sphingosine supplementation is that exogenous sphingosine is taken up and converted, as expected, to S1P which can directly inhibit the ceramide synthase 2 (Laviad et al., 2008) as one of the key enzymes in the de-novo pathway that converts dihydro-sphingosine to dihydro-ceramide. Therefore, more dihydro-sphingosine will be available for conversion to dihydro-S1P by the action of SKs.

Two studies have investigated the lipidomics in SPL knockout liver tissue and in isolated MEF cells (Bektas et al., 2010, and Gerl et al., 2016), which are included as references 80 and 81. In accordance to our observations, levels of S1P and sphingosine are increased, while dihydro-sphingosine, ceramides and hexosyl-ceramides are reduced. It is tempting to speculate that the accumulation of S1P creates a negative feedback on the de novo sphingolipid synthesis as a mechanism for the cell to re-establish a sphingolipid balance. This is now included in the discussion section (lines 245-248).

Indeed, we found that SK-1, but not SK-2, is downregulated on mRNA and protein level by SPL-kd. This is now included as new suppl. Fig. S8, and described in lines 245-248. In addition, SK-1 is strongly upregulated by TNFa, which may explain why dihydro-S1P accumulates so strongly in SPL-kd/TNFa cells.

Similarly, TNFalpha treatment of SPL-kd cells leads the up-regulation of sphinganine/dihydrosphingosine, the precursor of dihydroS1P (suppl. table). This could only be explained by an effect of TNFalpha on the de-novo sphingolipid metabolism independent of S1P lyase downregulation.

As already mentioned under point 1, TNFa strongly upregulates SK-1 (suppl. Fig. S8) and this, together with the SPL-kd-mediated dihydro-sphingosine accumulation would boost the synthesis of dihydro-S1P.

There is no evidence provided that S1P and dihydroS1P were not secreted when overproduced in the SPL-kd cells or as a result of sphingosine or TNFalpha treatments, therefore the conclusion that it is only intracellular effects of S1P or possibly dihydroS1P is incorrect. The S1P has been suggested to have an autocrine effects, which was not discussed as a possibility in this manuscript.

Indeed, we have addressed this point. As seen in suppl. Fig. S5, there was no change in the phospho-p42/p44-MAPK between control and SPL-kd. Since all S1PRs couple to p42/p44-MAPK activation, increased MAPK activation would have been expected if S1PRs had been activated by secreted S1P.

We have also incubated SPL-kd cells, which upregulate PECAM-1, with different S1P receptor antagonists against S1P1, S1P2, and S1P3. However, we could not see any downregulation in PECAM-1 expression.

Finally, with our ECISTM measurements we show that extracellular S1P by itself did not influence the barrier stability in either cell types (Fig. S9-12), suggesting that this mode of action is not important for the integrity of BBB-derived endothelial cells. In the inflammatory setting S1P did not have any effect on the already increased barrier stability of the SPL-kd.

From all these data, we conclude that secretion of S1P and autocrine action via S1P1,2 or 3 is not involved.

Labels on the blots of figs. 4, 5, and 7 need to be improved; It is not clear what the individual bands represent. Are those technical repeats and if yes of what? It is also not clear which bands are control bands and which are TNFalpha or TPA treated. This makes it difficult to interpret the results.

As suggested, we have changed the figure labels to make more clear which are the triplicates.

Round 2

Reviewer 1 Report

Dear authors

Thank you for your scientific reply and subsequently revising your manuscript. Although I'm still looking forward to seeing an animal model or in vivo data to translate your in vitro experiments, I have no further comments on your manuscript.

Reviewer 2 Report

The authors addressed my concerns.